# Oral health practices and oral hygiene status as indicators of suicidal ideation among adolescents in Southwest Nigeria

**Morenike Oluwatoyin Folayan**[1]*, **Maha El Tantawi**[2], **Olakunle Oginni**[3], **Elizabeth Oziegbe**[1], **Boladale Mapayi**[3], **Olaniyi Arowolo**[4], **Abiola Adetokunbo Adeniyi**[5], **Nadia A. Sam-Agudu**[6,7]

**1** Faculty of Dentistry, Obafemi Awolowo University, Ile-Ife, Osun State, Nigeria, **2** Faculty of Dentistry, Alexandria University, Alexandria, Alexandria Governorate, Egypt, **3** Department of Mental Health, Obafemi Awolowo University, Ile-Ife, Osun State, Nigeria, **4** Obafemi Awolowo University Teaching Hospitals' Complex, Ile-Ife, Osun State, Nigeria, **5** Faculty of Dentistry, University of British Columbia, Vancouver, British Columbia, Canada, **6** International Research Center of Excellence, Institute of Human Virology Nigeria, Abuja, Federal Capital Territory, Nigeria, **7** Institute of Human Virology, University of Maryland School of Medicine, Baltimore, Maryland, United States of America

* toyinukpong@yahoo.co.uk

**Data Availability Statement:** All relevant data are within the paper and its Supporting Information files.

## Abstract

### Background

Oral health is a less-recognized correlate of overall and mental wellbeing. This study aimed to assess the relationship between suicidal behavior (ideation and attempt) and oral health practices and status, and to determine the effect of sex on these associations among Nigerian adolescents.

### Methods

Household survey data were collected from 10 to 19-year-old adolescents in southwestern Nigeria. Dependent variables were daily tooth brushing, daily consumption of refined carbohydrates between meals, and oral hygiene status (measured by plaque index). The independent variable was lifetime suicidal ideation/attempt, dichotomized into 'yes' and 'never'. 'Daily tooth brushing' and 'daily consumption of refined carbohydrates between meals' were included in two separate logistic regression models, and 'oral hygiene status' was included in a linear regression model. The models were adjusted for sex, age, and socioeconomic status. The linear regression model was also adjusted for frequency of daily tooth-brushing and of consumption of refined carbohydrates between meals. Interactions between sex and suicidal ideation/suicide attempt in association with dependent variables were assessed. Significance was set at 5%.

### Results

We recruited 1,472 participants with mean age (standard deviation) of 14.6 (2.6) years. The mean plaque index was 0.84 (0.56), and 66 (4.5%) adolescents reported ever having suicidal ideation/attempt. Suicidal ideation/attempt was associated with significantly lower

**Funding:** The author(s) received no specific funding for this work.

**Competing interests:** The authors have declared that no competing interests exist.

likelihood of tooth brushing (OR = 0.48, 95% CI: 0.26, 0.91), higher likelihood of consuming refined carbohydrates between meals (OR = 2.30, 95% CI: 1.29, 4.10), and having poor oral hygiene (B = 0.18, 95% CI: 0.05, 0.32). Among males, suicidal ideation/attempt was associated with less likelihood of eating refined carbohydrates between meals (OR = 0.96, 95% CI: 0.35, 2.61). Conversely, it was associated with a significantly higher likelihood of this outcome (OR = 4.85, 95% CI: 2.23, 10.55) among females.

## Conclusion

The study findings suggest that poor tooth brushing habits and poor oral hygiene are indicators for risk of suicidal behavior for adolescents in Nigeria, while high sugar consumption may be an additional risk factor for adolescent females. These findings support the role of dental practitioners as members of healthcare teams responsible for screening, identifying and referring patients at risk for suicidal ideation/attempt.

## Introduction

Suicide was the second leading cause of death in 2017 among adolescents age 15 to 19 years globally [1]. Known risk factors include a family history of suicidal behavior; childhood and family adversity; mental disorders (untreated depression, substance use and psychotic disorders); exposure to stressors and adverse circumstances; sexual or gender minority status; previous suicide attempt; history of suicidal ideation [2–4].

A history of suicidal ideation/suicide attempt is associated with sugar addiction in adolescence [5]. While high sugar consumption has been associated with diminished cognitive function among the elderly [6], animal studies suggest that excessive consumption in adolescence is associated with overstimulation of the reward pathways [7], which may have negative effects on cognitive and emotion processing similar to the effects of substances of abuse [8]. High sugar consumption may result from neuroadaptations associated with depression [9,10], which is associated with increased risk for suicidal ideation/attempt and completed suicide [11]. High sugar consumption is also a risk factor for plaque accumulation and poor oral hygiene, which are risk factors for several oral and general health problems [12]. Depression leads to reductions in energy and self-esteem, which may lead to poor oral hygiene behaviors and health status [13,14]. Poor oral health is also a risk factor for depression [15]. Depression may therefore be a mediating factor between suicidal ideation/attempt and poor oral hygiene.

Studies have demonstrated associations between poor oral health, stress, depression, substance use disorders, and psychotic disorders [16]. The associations have been behavior-linked: mental health disorders are associated with poor self-care, including poor oral health care [17]. Few studies have examined the association between oral health and suicidal ideation: a study of Korean adults demonstrated no significant association [18], whereas a study of Aboriginal Australian young adults demonstrated an association [15].The disparate findings may indicate age, genetic, and/or social/environmental differences.

There are currently no studies on the relationship between suicidal behaviors and oral health status/behavior from African countries. In sub-Saharan Africa especially, adolescents are highly exposed to stressors and adverse childhood and family circumstances [19], and have a high prevalence of depression and other mental health problems [20,21], which are risk factors for suicidal ideation. Suicidal ideation/behavior is also common in the region: Nigeria has

a 12.4% to 17% prevalence of suicidal ideation and 7.8% for suicide attempts among adolescents [22,23]. These figures are higher than the reported 3.2% for suicidal ideation across several African countries: 1.0% for planned suicide, and 0.7% for attempted suicide [24]. Caries and poor oral hygiene, which are both consequences of high sugar consumption [25], are highly prevalent among adolescents across Africa [26] and in Nigeria [27].

This study is based on theories of hopelessness [28], which link suicidal ideation/attempts with psychopathological constructs such as depressive symptoms and hopelessness with their negative impact on agency thinking [29–31]. The hopelessness-low self-esteem-depression link is also associated with poor self-care [17], which includes oral health practices.

The study attempts to bridge a wide and longstanding gap in knowledge on interactions between oral health and mental health in Nigeria, the African region, and globally. The aim was to assess for associations between suicidal ideation/attempt and oral health practices/hygiene status, and to determine whether sex modified these associations. We hypothesized that there would be a negative association between suicidal ideation/attempt and oral health practices/oral hygiene status, whereby poor oral hygiene would be associated with higher-level risks of suicidal behavior, with sex likely modifying the association between these variables.

## Materials and methods

### Study design and study population

This is a secondary analysis of primary data collected to determine the association between the oral, mental, sexual, and reproductive health of adolescents' resident in Ife Central Local Government Area of Osun, State Ile-Ife, a semi-urban community in southwestern Nigeria. The data were collected through a household survey conducted during December 2018 and January 2019. Adolescents age 10–19 years old from whom parental consent/assent/individual informed consent was appropriately obtained were eligible to participate in the study. Adolescents who were critically ill and could not give independent responses to the study survey were excluded from study participation.

### Sample size and sampling technique

The minimum sample size for the study was calculated with the formula proposed by Araoye [32]. With a caries prevalence (proxy for oral hygiene status) of 13.9% among adolescents in the study community [33], a margin of error of 5%, and a confidence level of 95%, the minimum sample size was 1,323 adolescents. Adolescents were recruited with a multi-stage sampling technique. First, 70 of the 700 enumeration areas in Ife Central Local Government Area were sampled with the simple-random sampling technique. Next, every other household in the selected enumeration areas was identified as an eligible household. Finally, in each household, one adolescent who met the inclusion criteria was recruited for study participation. Whenever a household declined to participate, the next eligible household was substituted. Recruitment of participants continued until the minimum sample size for the study was reached.

### Data collection

Data were collected through personal interview with a structured questionnaire that had been used in previous studies on oral health in Nigeria (S1 File) [41]. The instrument was administered by trained field workers who were themselves young people with experience in collecting data for national surveys. The field workers and clinicians were trained on the study protocol, the use of the data collection tools, sample selection (including household listing and selection), and all other aspects of clinical and fieldwork. Data collected for each adolescent were

age, sex, and socioeconomic status. Age was determined in years as age at last birthday, while sex was determined as assigned sex at birth (male or female). Socio-economic status was measured with a proxy question that asked about average number of daily meals in the preceding month; the responses were categorized as 'cannot guarantee one meal per day', 'one meal per day', 'two meals per day', or 'three meals per day' [34].

**Oral health practices.**   Information was generated on the frequency of tooth brushing and consumption of refined carbohydrate between meals. Respondents were also asked to indicate the frequency of tooth brushing using the following response options–'irregularly or never', 'once a week', 'a few (2–3) times a week', 'once a day', and 'more than once a day'. Responses were further dichotomized into 'once a day or greater' and 'less than once a day'.

Respondents were also asked to indicate the frequency of consuming sugar-containing snacks or drinks between main meals according to the following options–'about three times a day or more', 'about twice a day', 'about once a day', 'occasionally', 'not every day', 'rarely', or 'never between meals'. Responses were dichotomized into 'three times a day or more' and 'less than three times a day'.

**Suicidal behavior assessment.**   Study participants' suicidal behavior was assessed with the Suicide Behaviour Questionnaire–Revised (SBQ-R). This is an easily administered 4-item tool that evaluates the frequency of past and likelihood of future suicidal thoughts and behaviors with responses scored on 3-7-point Likert scales [35,36]. The total scores were derived and used for analyses. The tool has adequate internal consistency in a population of patients with mental disorders in an outpatient setting (Cronbach's alpha 0.75) [37]. It has also been validated for use among undergraduate students in Nigeria (Cronbach's alpha 0.80) [38].

**Intra-oral examination.**   All participants had an oral examination conducted in their homes to determine the oral hygiene status. Each participant was examined sitting, under natural light, with sterile dental mirrors by trained dentists. The teeth were examined wet. Plaque Index [39] was used to determine the oral hygiene status. The Plaque Index score was based on six numerical determinations representing the amount of debris found on the surfaces of index permanent teeth 12, 16, 24, 32, 36, and 44. The mesial, distal, buccal, and lingual gingival areas of the index teeth are scored from 0 (no plaques) to 3 (abundance of soft matter within the gingival pocket and/or on the tooth and gingival margin). The mean score for each tooth is obtained and the mean score for the individual is obtained by adding the indices for each tooth and dividing by the number of teeth examined.

**Standardization of examiners.**   Clinical investigators were qualified dentists undergoing postgraduate residency training as pedodontists, who were calibrated on the study protocol and the clinical examination. Training was followed by practice on patients: each clinician examined and scored the adolescents for oral hygiene status as prescribed in the study protocol. Results were subjected to a Cohen's weighted kappa score analysis to determine intra- and inter-examiner variability. The intra- and inter-examiner Cohen's weighted kappa scores for the three dentists were all greater than 0.95.

## Data analysis

Descriptive statistics were calculated as means and standard deviations or frequencies and percentages. There were three separate dependent variables: brushing at least once daily (yes, no), consumption of refined carbohydrates between meals three times a day or more daily (yes, no), and oral hygiene status (mean plaque index). The first two dependent variables were included in two separate logistic regression models, and the third dependent variable was included in a linear regression model. The independent variable was lifetime suicidal ideation/ attempt dichotomized into 'yes' and 'never'. The models were adjusted for sex, age, and

socioeconomic status (measured using a proxy question that asked about average number of daily meals in the preceding month). These factors are associated with oral hygiene status and suicidal ideation/attempts in Nigeria [40,41]. The linear regression model, where oral hygiene status was a dependent variable, also adjusted for daily brushing and consumption of refined carbohydrates between meals. Interactions between sex and suicidal ideation/attempt in association with each of the three dependent variables were assessed. The p values for interaction were computed, and separate regression estimates were calculated for males and females. Odds rations/regression estimates and 95% confidence intervals (CI) were calculated. SPSS version 23.0 was used for statistical analysis. Significance was set at 5%.

### Ethical considerations

Ethical approval was obtained from the Health Research Ethics Committee of the Institute of Public Health at Obafemi Awolowo University in Ile-Ife, Nigeria. Written parental consent was obtained for all adolescents less than 18 years old per national ethics guidelines [42]. Written assent was additionally obtained for those 12 to 17 years of age. Adolescents aged 18 to 19 years provided written individual consent. Only adolescents that assented to study participation in addition to parental consent were enrolled in the study. All study participants received a gift of valued at approximately $1.00.

### Results

A total of 1,472 study participants were recruited (Table 1); age range of this cohort was 10 to 19 years. The mean age (standard deviation was 14.6 (2.6) years. There were 846 (57.5%) male participants, and 1,245 (84.6%) participants reported access to three meals a day in the preceding month. Most respondents (88.7%) reported brushing at least once daily, and 251 (17.1%) consumed refined carbohydrates between meals three or more time daily. The mean plaque index was 0.84 (0.56), and 202 (13.8%), 940 (64.0%), 310 (21.1%) and 17 (1.2%) participants had scores of 0, 1, 2, and 3 respectively. A total of 66 (4.5%) adolescents reported ever having suicidal ideation/attempts, 42 (2.8%) experienced suicidal ideation/attempts over the last 12 months, 20 (1.4%) reported they previously threatened to commit suicide, and 18 (1.2%) reported a likelihood of suicidal behavior in the future.

Table 2 lists the association between preventive oral health practices, oral hygiene status and suicidal ideation/attempt in the Nigerian adolescents. Being able to afford three meals a day was associated with significantly higher frequency of tooth brushing at least once a day (OR = 2.57, 95% CI: 1.74, 3.79), eating sugars (OR = 1.59, 95% CI: 1.02, 2.49), and better oral hygiene status (B = -0.10, 95%CI: -0.18, -0.01). Suicidal ideation/attempt was associated with significantly lower likelihood of tooth brushing (OR = 0.48, 95% CI: 0.26, 0.91) and significantly higher likelihood of consuming refined carbohydrates between meals daily (OR = 2.30, 95% CI: 1.29, 4.10) and poorer oral hygiene status (B = 0.18, 95% CI: 0.05, 0.32).

Table 3 illustrates the modifying effect of sex on the association between oral practices and suicidal ideation/attempt. Sex was significantly moderated the association between suicidal ideation/attempt and eating refined carbohydrates between meals daily (P = 0.01) but not the associations of suicidal ideation/attempt with tooth brushing (P = 0.72) or oral hygiene (P = 0.23). Among males, suicidal ideation/attempt was associated with a lower likelihood of eating refined carbohydrates between meals daily although this was not statistically significant (OR = 0.96, 95% CI: 0.35, 2.61); whereas, in females, suicidal ideation/attempt was associated with significantly higher likelihood of eating refined carbohydrates between meals daily (OR = 4.85, 95% CI: 2.23, 10.55).

**Table 1. Profiles, oral health practices and suicidal behaviors among Nigerian adolescents [N = 1,472].**

| Factors | Variables | Statistics |
|---|---|---|
| Sex | Male: n (%) | 846 (57.5) |
| | Female: n (%) | 626 (42.5) |
| Age | Mean (SD) | 14.6 (2.6) |
| Can afford three meals a day | Yes n (%) | 1245 (84.6) |
| | No n (%) | 227 (15.4) |
| Brush at least once a daily | Yes: n (%) | 1305 (88.7) |
| | No: n (%) | 167 (11.3) |
| Frequency of consumption of refined carbohydrates in-between-meals daily | three times a day or more n (%) | 251 (17.1) |
| | Less than that three times a day n (%) | 1221 (82.9) |
| Plaque index | Mean (SD) | 0.84 (0.56) |
| Suicidal ideation or attempt | Had thought n (%) | 66 (4.5) |
| | Never n (%) | 1406 (95.5) |
| Frequency of suicidal ideation over past 12 months | Had thought n (%) | 42 (2.8) |
| | Never n (%) | 1430 (97.2) |
| Threat of suicide attempt | Had thought n (%) | 20 (1.4) |
| | Never n (%) | 1452 (98.6) |
| Self-reported likelihood of suicidal behavior in the future | Had thought n (%) | 18 (1.2) |
| | Never n (%) | 1454 (98.8) |

## Discussion

The study findings indicate that suicidal behaviors were associated with poor oral health practices and oral hygiene status. Suicidal behaviors were also associated with less positive oral

**Table 2. Associations between brushing, sugar consumption, plaque accumulation, and suicidal ideation or attempt among Nigerian adolescents [N = 1472].**

| Variables | | Brushing at least once daily OR (95% CI) | Eating sugars daily OR (95% CI) | Oral hygiene status B (95% CI) |
|---|---|---|---|---|
| Sex | Male vs female | 0.87 (0.61, 1.22) | 1.05 (0.79, 1.41) | 0.04 (-0.02, 0.10) |
| Age | | 0.99 (0.93, 1.06) | 0.96 (0.90, 1.01) | -0.01 (-0.02, 0.004) |
| Afford three meals a day | | 2.57 (1.74, 3.79)* | 1.59 (1.02, 2.49)* | -0.10 (-0.18, -0.01)* |
| Brushing at least once daily | | - | - | -0.06 (-0.15, 0.03) |
| Eating refined carbohydrates in-between-meals daily | | | | -0.06 (-0.13, 0.02) |
| Suicidal ideation or attempt | | 0.48 (0.26, 0.91)* | 2.30 (1.29, 4.10)* | 0.18 (0.05, 0.32)* |

*OR: Odds ratio; B: Regression coefficient; CI: Confidence interval;*

*: *Statistically significant at P< 0.05.*

**Table 3. Modifying effect of sex on the association between tooth brushing, sugar consumption, plaque accumulation, and suicidal ideation/attempt [N = 1472].**

| Association between suicidal ideation/attempt and: | Male | Female | P for interaction |
|---|---|---|---|
| Brushing at least once daily [a]: OR (95% CI) | 0.45 (0.19, 1.05) | 0.56 (0.22, 1.47) | 0.72 |
| Eating refined carbohydrates in-between-meals daily [a]: OR (95% CI) | 0.96 (0.35, 2.61) | 4.85 (2.23, 10.55) | 0.01 |
| Plaque index [b]: B (95% CI) | -0.10 (-0.29, 0.09) | -0.24 (-0.44, -0.04) | 0.23 |

a: *Controlling for age and affording three meals a day.*

b: *Controlling for age, affording three meals a day, brushing at least once daily, and eating refined carbohydrates between meals daily.*

health practices such as insufficient tooth brushing, more negative practices such as daily between-meals consumption of refined carbohydrate, and with poor oral hygiene. In addition, there was sex difference in how suicidal ideation/attempt was associated with daily between-meal consumption of refined carbohydrates. A strong association was observed between suicidal ideation/attempt and daily consumption of refined carbohydrates between meals among female adolescents who had higher frequency of consumption of refined carbohydrate between meals and suicidal behaviors. Among males, in contrast, suicidal ideation/attempt was not significantly associated with any oral health behaviors. The study hypotheses were therefore validated.

This is the first study showing a relationship between oral health practices and suicidal ideation/attempt in a population of African adolescents. A strength of the study is the large sample size. Also, the household-based approach for sample recruitment renders the study findings generalizable to the study environment, and potentially possible to extrapolate the finding to other environments with similar population profiles. We however caution on extrapolation of findings to other cultures due to known influences of culture on mental health [43]. The study was conducted in a setting that was predominantly Yoruba, an ethnic group known to have very strong family orientation which is patho-protective [44]. The SBQ-R tool used to measure suicidal ideation/attempt had been validated among undergraduate students in the same study community. This strengthens the validity of study findings, as our study Cronbach's alpha was similar to that reported in the prior studies. Similarly, the instrument used to measure the oral hygiene behavior had been used in prior studies involving undergraduates [45], and preschool children [46] in the same study community.

The study is, however, limited by its cross-sectional design; thus, it can only suggest an association and not a cause-effect relationship. Also, there might be social desirability and recall bias for the self-reported items, which may lead to under-estimation of variables such as number of missed meals per day and over-estimation of factors such as frequency of tooth brushing. In addition, other mental problems, substance use issues and use of medication may confound the association between the observed variables and should be addressed in future studies. Despite this limitation, the study highlights important findings that should be explored further.

First, the association between suicidal tendencies and poor tooth brushing habits may not be associated with the experience of low self-esteem, which acts as an early entry into the suicidal process [47–49]. A prior study in India found an association between suicidal ideation/attempt and poor oral hygiene status, but not with tooth brushing frequency or tooth brushing duration [50]. The authors however reported an association between suicidal ideation/attempt and frequency of change of toothbrush: fewer individuals with suicidal ideation/attempt changed their toothbrush at least once in six months [50]. We found an association between suicidal tendencies and poor tooth brushing habits measured by frequency of tooth brushing. Low self-esteem is a strong predictor of poor tooth brushing habits [51], and may therefore be

the mediating factor between suicidal ideation/attempt and poor tooth brushing habits. This needs to be studied further.

Second, the association between suicidal tendencies and high daily between-meals consumption of refined carbohydrate may provide evidence to further substantiate prior findings on the association between sugar addiction and suicidal ideation/attempt [5], although there are multiple longitudinal studies showing an association of sugar consumption with an increased risk of depression [52–54]. Of concern is the risk for dental caries resulting from the high free-sugar consumption. Suicidal behavior is associated with poor treatment of dental caries [15], leading to pain, deteriorating quality of life, and risk for completed suicide [55,56].

Third, is the sex modifying role of the association between suicidal ideation/attempt and the daily consumption of refined carbohydrate between meals. This association, which was only found in females, is an important finding, as no prior study had highlighted the impact of sex on the sugar addiction and suicidal ideation/attempt relationship. The higher suicidal ideation/attempt for females has been associated with the higher risk of major depression in females, which is a predictive factor for more than half of all suicides [57]. The studies in Nigeria have consistently shown a female predilection for suicidal ideation/attempt [58,59]. Thus, considering the positive association between suicidal behaviors and depressive symptoms, which are higher among females, it is possible that the association between suicidality and high sugar consumption is mediated by depressive symptoms; however, future studies are needed to test this hypothesis. This finding may also point to the complexity of a sex relationship with suicidal ideation/attempt, as there are also suggestions of an underlying mechanism to sex difference in food choice and preferences including sweet tastes [58]. The possible role of culture in framing the relationship of sex and suicidal ideation/attempt [60] and moderating sex differences in taste preferences [61,62] needs to be explored further to explain the observed associations.

## Conclusion

Our findings suggest that poor tooth brushing habits and poor oral hygiene are indicators for the risk of suicidal behaviors for adolescents, and high sugar consumption may be an additional risk factor for adolescent females. These findings highlight the importance of dental practitioners as members of healthcare teams responsible for screening, identifying and referring at-risk adolescents in timely fashion.

## Supporting information

**S1 File. Study household survey questionnaire.**
(XLSX)

**S2 File.**
(DOCX)

## Acknowledgments

We acknowledge and thank the study participants for the contributions they made to generating new knowledge. Our appreciation also goes to field workers who collected study data.

## Author Contributions

**Conceptualization:** Morenike Oluwatoyin Folayan, Olakunle Oginni, Elizabeth Oziegbe, Boladale Mapayi, Abiola Adetokunbo Adeniyi, Nadia A. Sam-Agudu.

**Data curation:** Morenike Oluwatoyin Folayan, Olaniyi Arowolo.

**Formal analysis:** Maha El Tantawi.

**Investigation:** Morenike Oluwatoyin Folayan.

**Methodology:** Morenike Oluwatoyin Folayan, Maha El Tantawi, Olakunle Oginni, Elizabeth Oziegbe, Boladale Mapayi, Abiola Adetokunbo Adeniyi, Nadia A. Sam-Agudu.

**Project administration:** Morenike Oluwatoyin Folayan, Olaniyi Arowolo.

**Resources:** Morenike Oluwatoyin Folayan.

**Supervision:** Morenike Oluwatoyin Folayan.

**Validation:** Morenike Oluwatoyin Folayan, Maha El Tantawi, Olakunle Oginni.

**Writing – original draft:** Morenike Oluwatoyin Folayan, Maha El Tantawi.

**Writing – review & editing:** Morenike Oluwatoyin Folayan, Maha El Tantawi, Olakunle Oginni, Elizabeth Oziegbe, Boladale Mapayi, Olaniyi Arowolo, Abiola Adetokunbo Adeniyi, Nadia A. Sam-Agudu.

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
