## [Decision Letter · Decision Letter 0]

7 Jan 2021

PONE-D-20-34066

Oral health practices and oral hygiene status as indicators of suicidal ideation among Nigerian adolescents

PLOS ONE

Dear Dr. Folayan,

Thank you for submitting your manuscript to PLOS ONE. After careful consideration, we feel that it has merit but does not fully meet PLOS ONE’s publication criteria as it currently stands. Therefore, we invite you to submit a revised version of the manuscript that addresses the points raised during the review process.

We look forward to receiving your revised manuscript.

Kind regards,

Frédéric Denis, Ph.D.

Academic Editor

PLOS ONE

Journal Requirements:

Reviewers' comments:

Reviewer's Responses to Questions

**Comments to the Author**

1. Is the manuscript technically sound, and do the data support the conclusions?

Reviewer #1: Yes

Reviewer #2: Yes

2. Has the statistical analysis been performed appropriately and rigorously? 

Reviewer #1: I Don't Know

Reviewer #2: Yes

3. Have the authors made all data underlying the findings in their manuscript fully available?

Reviewer #1: Yes

Reviewer #2: Yes

4. Is the manuscript presented in an intelligible fashion and written in standard English?

Reviewer #1: Yes

Reviewer #2: Yes

5. Review Comments to the Author

Reviewer #1: This is an extremely relevant subject with an impressive sample; it deserves publication and requires only minor adjustments. Below are suggestions to improve the article. Congratulations on your beautiful study!

Introduction

1. I suggest removing the excerpt about 2014 from the first paragraph, leaving only the information about 2017. It is a bit repetitive.

2. Rephrase the following excerpt in a more clear and objective way - “Suicidal ideation/attempt may also increase the risk for poor toothbrushing and plaque accumulation through a common link - depression. Depression leads to reduction in energy, self-worth, and esteem, which may lead to poor toothbrushing, plaque accumulation, and poor oral hygiene. Poor oral health is also a risk factor for depression.”

3. The introduction section is too long, it requires more objectivity. In addition, the results fill a gap not only in the study’s region but a worldwide gap. We know that studies like these, with a large sample size, are scarce.

Methods and Results

1. Could the studied community have influenced the results?

2. Were mental health, medication and substance (drugs) use also investigated? Could these factors influence the results?

3. Why were the results not compared with a control group? Adolescents with no intention to commit or no attempted suicide? We know that adolescence is a period when oral hygiene decreases and sugar consumption increases.

4. The article mentions caries evaluation in the methods section. How was it accomplished? Where is the result? Was periodontal disease evaluated? Oral injuries?

5. The numerical values can be better described, at least by presenting minimum, maximum and median values as well.

6. The use of graphs showing associations and trends can further enhance the article.

Discussion

1. The discussion section could extrapolate a little more by using data from the literature, as well as touch a little more on the visible plaque index and on the questionnaire used to assess suicide.

Conclusion – this section is objective and impactful!

Reviewer #2: The reviewer would like to thank the authors for their interesting study. However some minor comments should be considered to improve the manuscript.

1- The abstract is very long and does not summarize the study. Please provide a short abstract that presents only the most important information of the investigation.

2- The authors discussed the limitations of the study in the first part of the discussion. It would be much better if the authors created a paragraph at the end of the discussion about the limitations. This makes it easier for the reader to understand all aspects and limitations discussed.

3- The authors provided a clear discussion of the relationship between high sugar consumption, depression and the suicidal rate. Nevertheless, some points need to be further discussed, as the role of the socioeconomical status. Is the number of meals per day also an effective instrument to conclude the socioeconomical status of the participants? This also needs to be discussed further.

4- Are there similar studies in other countries, maybe outside Africa? Please discuss and compare if available.

5- Formatting mistakes should be corrected

6. PLOS authors have the option to publish the peer review history of their article (what does this mean?). If published, this will include your full peer review and any attached files.

Reviewer #1: **Yes: **bruna lavinas sayed picciani

Reviewer #2: No

---

## [Author Response · Author response to Decision Letter 0]

28 Jan 2021

Response to Reviewers

PLOS One Article PONE-D-20-34066

Oral health practices and oral hygiene status as indicators of suicidal ideation among Nigerian adolescents

15 January, 2021

Journal Requirements:

Response: We have thoroughly re-examined and revised our formatting in line with PLOS One’s style requirements. Heading font styles and sizes, reference citations and styles and supplementary file naming have all been checked and corrected. Thank you for bringing this to our attention.

Response: Thanks. We have uploaded the anonymised data set used for this study along with the revised manuscript. 

Response: In text citation (S1 File) and a Supporting Information file section for our supplemental file have been added accordingly. 

Review Comments to the Author

Reviewer #1: 

This is an extremely relevant subject with an impressive sample; it deserves publication and requires only minor adjustments. Below are suggestions to improve the article. Congratulations on your beautiful study!

Response: Thank you for the kind words!

Introduction

1. I suggest removing the excerpt about 2014 from the first paragraph, leaving only the information about 2017. It is a bit repetitive. 

Response: Thank you for bringing this to our attention; the excerpt has been removed.

2. Rephrase the following excerpt in a more clear and objective way - “Suicidal ideation/attempt may also increase the risk for poor toothbrushing and plaque accumulation through a common link - depression. Depression leads to reduction in energy, self-worth, and esteem, which may lead to poor toothbrushing, plaque accumulation, and poor oral hygiene. Poor oral health is also a risk factor for depression.” 

Response. Sentence revised to read: Depression leads to reductions in energy and self-esteem, which may lead to poor oral hygiene behaviors and health status [13, 14]. Poor oral health is also a risk factor for depression [15]. Depression may therefore be a mediating factor between suicidal ideation/attempt and poor oral hygiene.

3. The introduction section is too long, it requires more objectivity. In addition, the results fill a gap not only in the study’s region but a worldwide gap. We know that studies like these, with a large sample size, are scarce.

Response: Thank you for the helpful critique. We have reduced the length of the Introduction section; it is now less than 600 words (574 words to be exact). We have also made edits to portray more objectivity and to highlight that our findings are potentially bridging national, regional and global gaps in oral/mental health research.

Methods and Results

1. Could the studied community have influenced the results? 

Response: This is a possibility. We have addressed this in the manuscript and cautioned on the generalisability of the study finding because of cultural differences in communities. Please see the second paragraph in the Discussion section of the manuscript.

2. Were mental health, medication and substance (drugs) use also investigated? Could these factors influence the results? 

Response: We do agree that mental health, medication and substance use may influence the results. They were not, however, included in this study. We added a narrative in Discussion/ Limitations about this and called for future studies to address this gap in knowledge.

3. Why were the results not compared with a control group? Adolescents with no intention to commit or no attempted suicide? We know that adolescence is a period when oral hygiene decreases and sugar consumption increases. 

Response: The objective for this study was to identify oral health practices and behaviours associated with suicidal ideation/attempt. We do acknowledge that a study identifying differences in oral health practices and behaviours in adolescents with and without suicidal ideation is a good mini study. We will conduct this analysis and publish as a separate manuscript

4. The article mentions caries evaluation in the methods section. How was it accomplished? Where is the result? Was periodontal disease evaluated? Oral injuries? 

Response: Thanks for picking up this gap in the methodology. This study only determined the association between suicidal ideation/attempt and oral hygiene status. The oral hygiene status is a risk factor for caries, periodontal disease and other general health problems. We have adjusted the methods to remove ambiguity regarding caries evaluation. Unfortunately, we did not have data for periodontal disease and oral injuries. 

5. The numerical values can be better described, at least by presenting minimum, maximum and median values as well. 

Response: The study cohort’s age range (10 to 19 years) as well as the range for plaque scores have now been included in first paragraph of the Results section’s narrative. We have provided range for the data it is feasible to provide these values for

6. The use of graphs showing associations and trends can further enhance the article.

Response: We used logistic regression for two dependent variables (brushing at least once daily and consumption of refined carbohydrates in between meals) and this displays no plots for associations, trends or interactions. Also, in the linear regression, the independent variable of interest was suicidal ideation or attempt which is a binary variable for which no interaction plot can be produced. The only alternative left was to use clustered bar charts for the association between suicidal ideation or attempt and the two categorical dependent variables (brushing and consumption of refined carbohydrates) after splitting the sample by sex. If these were used, they would show bivariate associations without the adjustment of regression analyses. This would distort the results and mislead readers. Because of this, we chose not to use graphs.

Discussion

1. The discussion section could extrapolate a little more by using data from the literature, as well as touch a little more on the visible plaque index and on the questionnaire used to assess suicide. 

Response: We discussed the use of merits and demerits of use of the questionnaire used to assess suicide as oral hygiene behavior in the second paragraph of the discussion section. We have also included some references and Discussion narrative comparing the study findings with other findings. Please see references 43-46.

2. Conclusion – this section is objective and impactful! 

Response: Thank you for the kind assessment.

Reviewer #2: 

The reviewer would like to thank the authors for their interesting study. However some minor comments should be considered to improve the manuscript.

1. The abstract is very long and does not summarize the study. Please provide a short abstract that presents only the most important information of the investigation. 

Response: This has been revised and word number reduced accordingly; it is now 348 words long, and we only include salient results.

2. The authors discussed the limitations of the study in the first part of the discussion. It would be much better if the authors created a paragraph at the end of the discussion about the limitations. This makes it easier for the reader to understand all aspects and limitations discussed. 

Response: We followed the STROBE guidelines for cross sectional studies; it requires the strength and limitations of the study be written up immediately after the first paragraph that gives a summary of the results. 

3. The authors provided a clear discussion of the relationship between high sugar consumption, depression and the suicidal rate. Nevertheless, some points need to be further discussed, as the role of the socioeconomical status. Is the number of meals per day also an effective instrument to conclude the socioeconomical status of the participants? This also needs to be discussed further. 

Response: Thank you for this query. There are previous studies to demonstrate that the number of meals accessed per day is an indication of the socioeconomic status for children in Nigeria. Please see Sodipo MA et al. Influence of socio-economic status on intake of lunch by school age children. FUTA J. Res. Sci. 2017; 13 (1): 129-136

4. Are there similar studies in other countries, maybe outside Africa? Please discuss and compare if available. 

Response: We have improved the Discussion by making references to other studies. There are precious few studies to compare with though. Please see references 50, 52-54.

5. Formatting mistakes should be corrected. 

Response: Thanks for highlighting this. We have effected extensive formatting edits: throughout the manuscript (particularly Title Page, heading fonts) and in the References and Supporting Information sections. We trust that all formatting mistakes have been taken care of.

---

## [Editor Report · Decision Letter 1]

1 Feb 2021

Oral health practices and oral hygiene status as indicators of suicidal ideation among adolescents in Southwest Nigeria

PONE-D-20-34066R1

Dear Dr. Folayan,

We’re pleased to inform you that your manuscript has been judged scientifically suitable for publication and will be formally accepted for publication once it meets all outstanding technical requirements.

Kind regards,

Frédéric Denis, Ph.D.

Academic Editor

PLOS ONE
---

## [Editor Report · Acceptance letter]

16 Feb 2021

PONE-D-20-34066R1 

Oral health practices and oral hygiene status as indicators of suicidal ideation among adolescents in Southwest Nigeria 

Dear Dr. Folayan:

I'm pleased to inform you that your manuscript has been deemed suitable for publication in PLOS ONE. Congratulations! Your manuscript is now with our production department. 

Kind regards, 

on behalf of

Dr. Frédéric Denis 

Academic Editor

PLOS ONE